# Watch and Match: Supercharging Imitation with Regularized Optimal Transport

**Siddhant Haldar**[1]  **Vaibhav Mathur**  **Denis Yarats**  **Lerrel Pinto**

New York University

[rot-robot.github.io](rot-robot.github.io)

**Abstract:**

Imitation learning holds tremendous promise in learning policies efficiently for complex decision making problems. Current state-of-the-art algorithms often use inverse reinforcement learning (IRL), where given a set of expert demonstrations, an agent alternatively infers a reward function and the associated optimal policy. However, such IRL approaches often require substantial online interactions for complex control problems. In this work, we present Regularized Optimal Transport (ROT), a new imitation learning algorithm that builds on recent advances in optimal transport based trajectory-matching. Our key technical insight is that adaptively combining trajectory-matching rewards with behavior cloning can significantly accelerate imitation even with only a few demonstrations. Our experiments on 20 visual control tasks across the DeepMind Control Suite, the OpenAI Robotics Suite, and the Meta-World Benchmark demonstrate an average of $7.8\times$ faster imitation to reach $90\%$ of expert performance compared to prior state-of-the-art methods. On real-world robotic manipulation, with just one demonstration and an hour of online training, ROT achieves an average success rate of 90.1% across 14 tasks.

**Keywords:** Imitation Learning, Manipulation, Robotics

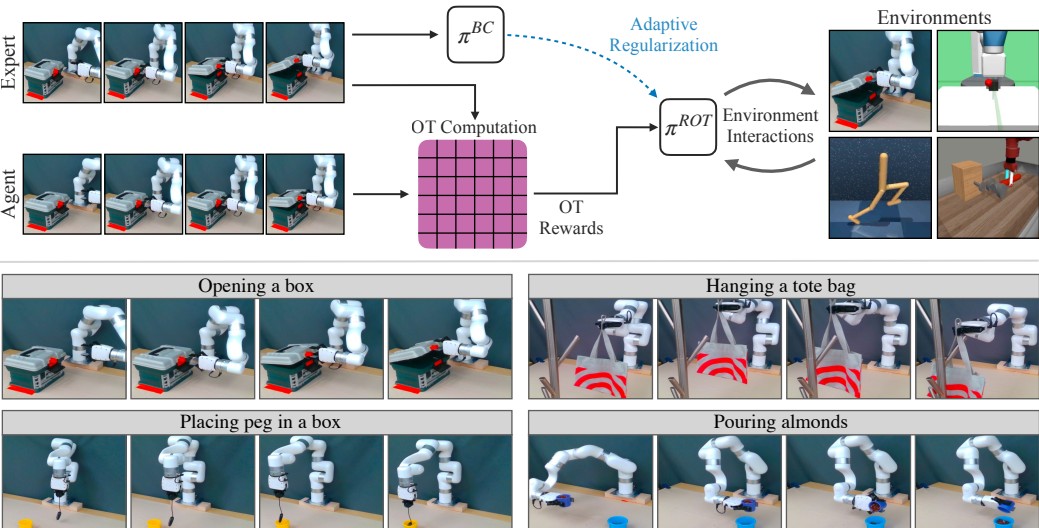

Figure 1: **(Top)** Regularized Optimal Transport (ROT) is a new imitation learning algorithm that adaptively combines offline behavior cloning with online trajectory-matching based rewards. This enables significantly faster imitation across a variety of simulated and real robotics tasks, while being compatible with high-dimensional visual observation. **(Bottom)** On our xArm robot, ROT can learn visual policies with only a single human demonstration and under an hour of online training.

---

[1]Correspondence to: siddhanthaldar@nyu.edu

6th Conference on Robot Learning (CoRL 2022), Auckland, New Zealand.

# 1 Introduction

Imitation Learning (IL) [1, 2, 3] has a rich history that can be categorized across two broad paradigms, Behavior Cloning (BC) [1] and Inverse Reinforcement Learning (IRL) [4]. BC uses supervised learning to obtain a policy that maximizes the likelihood of taking the demonstrated action given an observation in the demonstration. While this allows for training without online interactions, it suffers from distributional mismatch during online rollouts [5]. IRL, on the other hand, infers the underlying reward function from the demonstrated trajectories before employing RL to optimize a policy through online environment rollouts. This results in a policy that can robustly solve demonstrated tasks even in the absence of task-specific rewards [6, 7].

Although powerful, IRL methods suffer from a significant drawback – they require numerous expensive online interactions with the environment. There are three reasons for this: (a) the inferred reward function is often highly non-stationary, which compromises the learning of the associated behavior policy [7]; (b) even when the rewards are stationary, policy learning still requires effective exploration to maximize rewards [8]; and (c) when strong priors such as pretraining with BC are applied to accelerate policy learning, ensuing updates to the policy cause a distribution shift that destabilizes training [9, 10]. Combined, these issues manifest themselves on empirical benchmarks, where IRL methods have poor efficiency compared to vanilla RL methods on hard control tasks [11].

In this work, we present Regularized Optimal Transport (ROT) for imitation learning, a new method that is conceptually simple, compatible with high-dimensional observations, and requires minimal additional hyperparameters compared to standard IRL approaches. In order to address the challenge of reward non-stationarity in IRL, ROT builds upon recent advances in using Optimal Transport (OT) [12, 13, 11] for reward computation that use non-parametric trajectory-matching functions. To alleviate the challenge of exploration, we pretrain the IRL behavior policy using BC on the expert demonstrations. This reduces the need for our imitation agent to explore from scratch.

However, even with OT-based reward computation and pretrained policies, we only obtain marginal gains in empirical performance. The reason for this is that the high-variance of IRL policy gradients [14, 15] often wipe away the progress made by the offline BC pretraining. This phenomenon has been observed in both online RL [16] and offline RL [9] methods. Inspired by solutions presented in these works, we stabilize the online learning process by regularizing the IRL policy to stay close to the pretrained BC policy. To enable this, we develop a new adaptive weighing scheme called soft Q-filtering that automatically sets the regularization – prioritizing staying close to the BC policy in the beginning of training and prioritizing exploration later on. In contrast to prior policy regularization schemes [16, 17], soft Q-filtering does not require hand-specification of decay schedules.

To demonstrate the effectiveness of ROT, we run extensive experiments on 20 simulated tasks across DM Control [18], OpenAI Robotics [19], and Meta-world [20], and 14 robotic manipulation tasks on an xArm (see Fig. 1). Our main findings are summarized below.

1. ROT outperforms prior state-of-the-art imitation methods, reaching $90\%$ of expert performance $7.8\times$ faster than our strongest baselines on simulated visual control benchmarks.

2. On real-world tasks, with a single human demonstration and an hour of training, ROT achieves an average success rate of 90.1% with randomized robot initialization and image observations. This is significantly higher than behavior cloning (36.1%) and adversarial IRL (14.6%).

3. ROT exceeds the performance of state-of-the-art RL trained with rewards, while coming close to methods that augment RL with demonstrations (Section 4.5 & Appendix H.3). Unlike standard RL methods, ROT does not require hand-specification of the reward function.

4. Ablation studies demonstrate the importance of every component in ROT, particularly the role that soft Q-filtering plays in stabilizing training and the need for OT-based rewards during online learning (Section 4.4 & Appendix H.4).

Open-source code and demonstration data has been publicly released on our project website. Videos of our trained policies can be seen here: `rot-robot.github.io`.

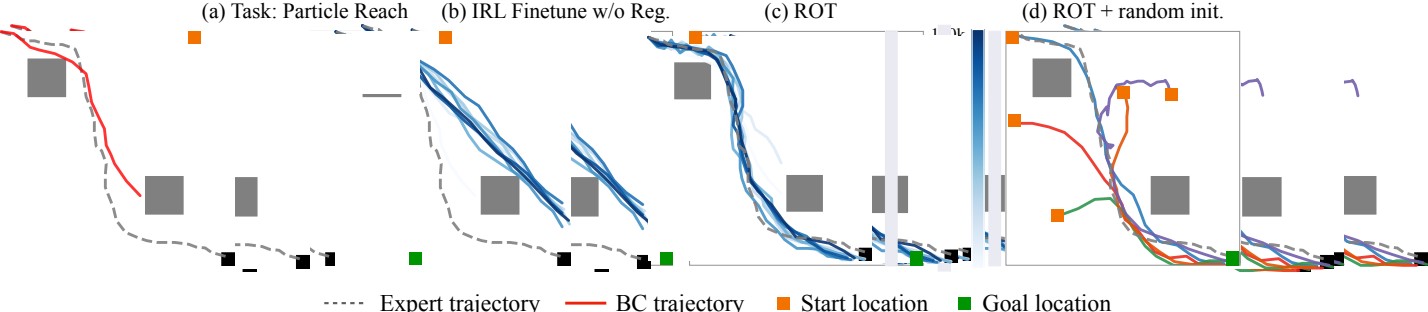

Figure 2: Given a single demonstration to avoid the grey obstacle and reach the goal location, BC is unable to solve the task (a). Finetuning from this BC policy with OT-based reward also fails to solve the task (b). ROT, with adaptive regularization of OT-based IRL with BC successfully solves the task (c). Even when the ROT agent is initialized randomly, it is able to solve the task (d).

## 2 Background

Before describing our method in detail, we provide a brief background to imitation learning with optimal transport, which serves as the backbone of our method. Formalism related to RL follows the convention in prior work [8, 11] and is described in Appendix A.

**Imitation Learning with Optimal Transport (OT)**   The goal of imitation learning is to learn a behavior policy $\pi^b$ given access to either the expert policy $\pi^e$ or trajectories derived from the expert policy $\mathcal{T}^e$. While there are a multitude of settings with differing levels of access to the expert [21], our work operates in the setting where the agent only has access to observation-based trajectories, i.e. $\mathcal{T}^e \equiv \{(o_t, a_t)_{t=1}^T\}_{n=1}^N$. Here $N$ and $T$ denotes the number of trajectory rollouts and episode timesteps respectively. Inverse Reinforcement Learning (IRL) [4, 22] tackles the IL problem by inferring the reward function $r^e$ based on expert trajectories $\mathcal{T}^e$. Then given the inferred reward $r^e$, policy optimization is used to derive the behavior policy $\pi^b$. To compute $r^e$, a new line of OT-based approaches for IL [12, 13, 11] have been proposed. Intuitively, the closeness between expert trajectories $\mathcal{T}^e$ and behavior trajectories $\mathcal{T}^b$ can be computed by measuring the optimal transport of probability mass from $\mathcal{T}^b \rightarrow \mathcal{T}^e$. Thus, given a cost matrix $C_{t,t'} = c(o_t^b, o_{t'}^e)$ and the optimal alignment $\mu^*$ between a behavior trajectory $o^b$ and and expert trajectory $o^e$, a reward signal for each observation can be computed using the equation:

$$r^{OT}(o_t^b) = -\sum_{t'=1}^{T} C_{t,t'}\mu_{t,t'}^*$$ 

(1)

A detailed account of the OT formulation has been provided in Appendix A.

**Actor-Critic based reward maximization**   Given rewards obtained through OT computation, efficient maximization of the reward can be achieved through off-policy learning [7]. In this work, we use Deep Deterministic Policy Gradient (DDPG) [23] as our base RL optimizer which is an actor-critic algorithm that concurrently learns a deterministic policy $\pi_\phi$ and a Q-function $Q_\theta$. However, instead of minimizing a one step Bellman residual in vanilla DDPG, we use the recent n-step version of DDPG from Yarats et al. [8] that achieves high performance on visual control problems.

## 3 Regularized Optimal Transport

A fundamental challenge in imitation learning is to balance the ability to mimic demonstrated actions along with the ability to recover from states outside the distribution of demonstrated states. Behavior Cloning (BC) specializes in mimicking demonstrated actions through supervised learning, while Inverse Reinforcement Learning (IRL) specializes in obtaining policies that can recover from arbitrary states. Regularized Optimal Transport (ROT) combines the best of both worlds by adaptively combining the two objectives. The challenges in online finetuning from a pretrained policy have been

described in Fig. 2, with more details provided in Appendix B.1. ROT operates in two phases. In the first phase, a randomly initialized policy is trained using the BC objective on expert demonstrated data. This 'BC-pretrained' policy then serves as an initialization for the second phase. In the second phase, the policy is allowed access to the environment where it can train using an IRL objective. To accelerate the IRL training, the BC loss is added to the objective with an adaptive weight. Details of each component are described below, with additional algorithmic details in Appendix C.

## 3.1 Phase 1: BC Pretraining

BC corresponds to solving the maximum likelihood problem shown in Eq. 2. Here $\mathcal{T}^e$ refers to expert demonstrations. When parameterized by a normal distribution with fixed variance, the objective can be framed as a regression problem where, given inputs $s^e$, $\pi^{BC}$ needs to output $a^e$.

$$\mathcal{L}^{BC} = \mathbb{E}_{(s^e, a^e) \sim \mathcal{T}^e} \| a^e - \pi^{BC}(s^e) \|^2 \tag{2}$$

After training, it enables $\pi^{BC}$ to mimic the actions corresponding to the observations seen in the demonstrations. However, during rollouts in an environment, small errors in action prediction can lead to the agent visiting states not seen in the demonstrations [5]. This distributional mismatch often causes $\pi^{BC}$ to fail on empirical benchmarks [16, 11] (see Fig. **??** (a) in Appendix B).

## 3.2 Phase 2: Online Finetuning with IRL

Given a pretrained $\pi^{BC}$ model, we now begin online 'finetuning' of the policy $\pi^b \equiv \pi^{ROT}$ in the environment. Since we are operating without explicit task rewards, we use rewards obtained through OT-based trajectory matching, which is described in Section 2. These OT-based rewards $r^{OT}$ enable the use of standard RL optimizers to maximize cumulative reward from $\pi^b \equiv \pi^{ROT}$. In this work we use n-step DDPG [23], a deterministic actor-critic based method that provides high-performance in continuous control [8].

**Finetuning with Regularization**    $\pi^{BC}$ is susceptible to distribution shift due to accumulation of errors during online rollouts [5] and directly finetuning $\pi^{BC}$ also leads to subpar performance (refer to Fig. 2). To address this, we build upon prior work in guided RL [16] and offline RL [9], and regularize the training of $\pi^{ROT}$ by combining it with a BC loss as seen in Eq. 3.

$$\pi^{ROT} = \underset{\pi}{\operatorname{argmax}} \left[ (1 - \lambda(\pi))) \mathbb{E}_{(s,a) \sim \mathcal{D}_\beta}[Q(s,a)] - \alpha \lambda(\pi) \mathbb{E}_{(s^e, a^e) \sim \mathcal{T}^e} \| a^e - \pi(s^e) \|^2 \right] \tag{3}$$

Here, $Q(s, a)$ represents the Q-value from the critic which is optimized using OT-based rewards during the actor-critic policy optimization. $\alpha$ is a fixed weight, while $\lambda(\pi)$ is a policy-dependent adaptive weight that controls the contributions of the two loss terms. $\mathcal{D}_\beta$ refers to the replay buffer for online rollouts.

**Adaptive Regularization with Soft Q-filtering**    While prior work [16, 17] use hand-tuned schedules for $\lambda(\pi)$, we propose a new adaptive scheme that removes the need for tuning. This is done by comparing the performance of the current policy $\pi^{ROT}$ and the pretrained policy $\pi^{BC}$ on a batch of data sampled from the replay buffer for online rollouts $\mathcal{D}_\beta$. More precisely, given a behavior policy $\pi^{BC}(s)$, the current policy $\pi^{ROT}(s)$, the Q-function $Q(s, a)$ and the replay buffer $\mathcal{D}_e$, we set $\lambda$ as:

$$\lambda(\pi^{ROT}) = \mathbb{E}_{(s, \cdot) \sim \mathcal{D}_\beta} \left[ \mathbb{1}_{Q(s, \pi^{BC}(s)) > Q(s, \pi^{ROT}(s))} \right] \tag{4}$$

The strength of the BC regularization hence depends on the performance of the current policy with respect to the behavior policy. This filtering strategy is inspired by Nair et al. [24], where instead of a binary hard assignment we use a soft continuous weight. Experimental comparisons with hand-tuned decay strategies are presented in Section 4.4.

**Considerations for image-based observations**    Since we are interested in using ROT with high-dimensional visual observations, additional machinery is required to ensure compatibility. Following prior work in image-based RL and imitation [8, 11], we perform data augmentations on visual observations and then feed it into a CNN encoder. Similar to Cohen et al. [11], we use a target

encoder with Polyak averaging to obtain representations for OT reward computation. This is necessary to reduce the non-stationarity caused by learning the encoder alongside the ROT imitation process. Further implementation details and the training procedure can be found in Appendix C.

## 4 Experiments

Our experiments are designed to answer the following questions: (a) How efficient is ROT for imitation learning? (b) How does ROT perform on real-world tasks? (c) How important is the choice of IRL method in ROT? (d) Does soft Q-filtering improve imitation? (e) How does ROT compare to standard reward-based RL? Additional results and analysis have been provided in Appendix H.

**Simulated tasks**  We experiment with 10 tasks from the DeepMind Control suite [18, 25], 3 tasks from the OpenAI Robotics suite [26], and 7 tasks from the Meta-world suite [27]. For DeepMind Control tasks, we train expert policies using DrQ-v2 [8] and collect 10 demonstrations for each task using this policy. For OpenAI Robotics tasks, we train a state-based DrQ-v2 with hindsight experience replay [28] and collect 50 demonstrations for each task. For Meta-world tasks, we use a single hard-coded expert demonstration from their open-source implementation [27]. Full environment details can be found in Appendix D and details about the variations in demonstrations and initialization conditions can be found in Appendix E.

**Robot tasks**  Our real world setup for each of the 14 manipulation tasks can be seen in Fig. 4. We use an Ufactory xArm 7 robot with a xArm Gripper as the robot platform for our real world experiments. However, our method is agnostic to the specific robot hardware. The observations are RGB images from a fixed camera. In this setup, we only use a single expert demonstration collected by a human operator with a joystick and limit the online training to a fixed period of 1 hour. Descriptions of each task and the evaluation procedure is in Appendix F.

**Primary baselines**  We compare ROT with baselines against several prominent imitation learning methods. While a full description of our baselines are in Appendix G, a brief description of the two strongest ones are as follows:

1. **Adversarial IRL (DAC):** Discriminator Actor Critic [7] is a state-of-the-art adversarial imitation learning method [6, 29, 7]. DAC outperforms prior work such as GAIL [6] and AIRL [30], and thus it serves as our primary adversarial imitation baseline.
2. **Trajectory-matching IRL (OT):** Sinkhorn Imitation Learning [12, 13] is a state-of-the-art trajectory-matching imitation learning method [31] that approximates OT matching through the Sinkhorn Knopp algorithm [32, 33]. Since ROT is derived from similar OT-based foundations, we use SIL as our primary state-matching imitation baseline.

### 4.1 How efficient is ROT for imitation learning?

Performance of ROT for image-based imitation is depicted on select environments in Fig. 3. On all but one task, ROT trains significantly faster than prior work. To reach 90% of expert performance, ROT is on average $8.7\times$ faster on DeepMind Control tasks, $2.1\times$ faster on OpenAI Robotics tasks, and $8.9\times$ faster on Meta-world tasks. We also find that the improvements of ROT are most apparent on the harder tasks, which are in rightmost column of Fig. 3. Appendix H.1 shows results on all 20 simulated tasks, along with experiments that exhibit similar improvements in state-based settings.

### 4.2 How does ROT perform on real-world tasks?

We devise a set of 14 manipulation tasks on our xArm robot to compare the performance of ROT with BC and our strongest baseline RDAC, an adversarial IRL method that combines DAC [7] with our pretraining and regularization scheme. The BC policy is trained using supervised learning on a single expert demonstration collected by a human operator. ROT and RDAC finetune the pretrained BC policy through 1 hour of online training, which amounts to $\sim$ 6k environment steps. Since there is just one demonstration, our tasks are designed to have random initializations but fixed goals. Note that a single demonstration only demonstrates solving the tasks from one initial condition. Evaluation results across 20 different initial conditions can be seen in Fig. 4. We observe that ROT

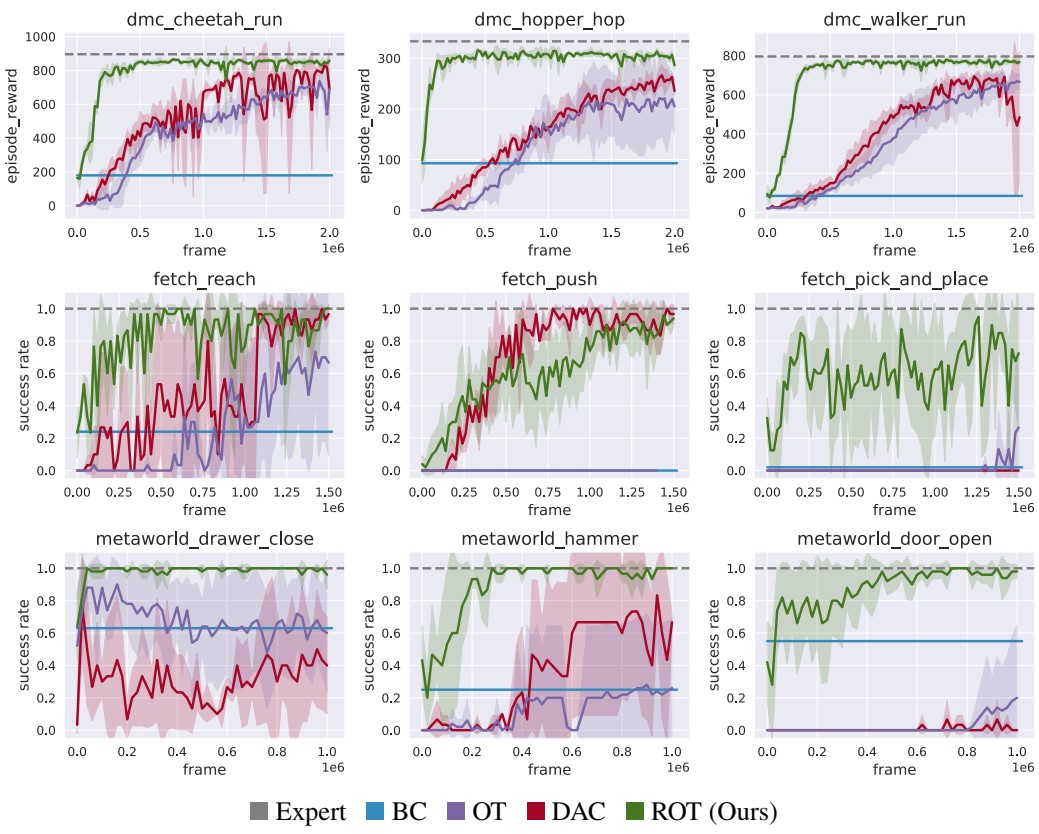

Figure 3: Pixel-based continuous control learning on 9 selected environments. Shaded region represents ±1 standard deviation across 5 seeds. We notice that ROT is significantly more sample efficient compared to prior work.

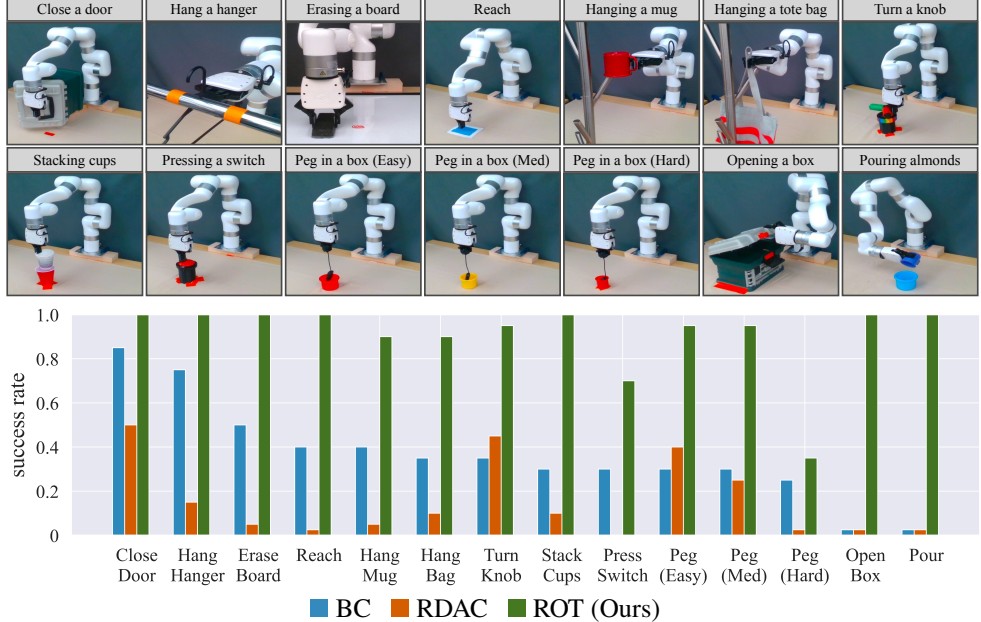

Figure 4: **(Top)** ROT is evaluated on a set of 14 robotic manipulation tasks. **(Bottom)** Success rates for each task is computed by running 20 trajectories from varying initial conditions on the robot.

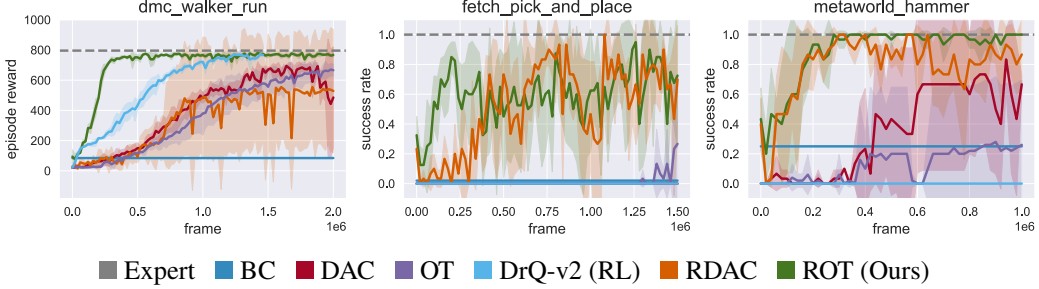

Figure 5: Ablation analysis on the choice of base IRL method. We find that although adversarial methods benefit from regularized BC, the gains seen are smaller compared to ROT. Here, we also see that ROT can outperform plain RL that requires explicit task-rewards.

has an average success rate of 90.1% over 20 evaluation trajectories across all tasks as compared to 36.1% for BC and 14.6% for RDAC. The poor performance of BC can be attributed to distributional mismatch due to accumulation of error in online rollouts and different initial conditions. The poor performance of RDAC can be attributed to slow learning during the initial phase of training. More detailed evaluations of RDAC on simulated environments is present in Sec. 4.3.

### 4.3 How important is the choice of IRL method in ROT?

In ROT, we build on OT-based IRL instead of adversarial IRL. This is because adversarial IRL methods require iterative reward learning, which produces a highly non-stationary reward function for policy optimization. In Fig. 5, we compare ROT with adversarial IRL methods that use our pretraining and adaptive BC regularization technique (RDAC). We find that our soft Q-filtering method does improve prior state-of-the-art adversarial IRL (RDAC vs. DAC in Fig. 5). However, our OT-based approach (ROT) is more stable and on average leads to more efficient learning.

### 4.4 Does soft Q-filtering improve imitation?

To understand the importance of soft Q-filtering, we compare ROT against two variants of our proposed regularization scheme: (a) A tuned fixed BC regularization weight (ignoring $\lambda(\pi)$ in Eq. 3); (b) A carefully designed linear-decay schedule for $\lambda(\pi)$, where it varies from $1.0$ to $0.0$ in the first 20k environment steps [16]. As demonstrated in Fig. 6 (and Appendix H.2), ROT is on par and in some cases exceeds the efficiency of a hand-tuned decay schedule, while not having to hand-tune its regularization weights. We hypothesize this

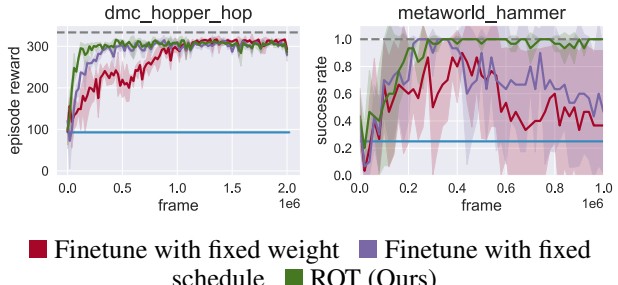

Figure 6: Effect of various BC regularization schemes compared with our adaptive soft-Q filtering regularization.

improvement is primarily due to the better stability of adaptive weighing as seen in the significantly smaller standard deviation on the Meta-world tasks.

### 4.5 How does ROT compare to standard reward-based RL?

We compare the performance of ROT against DrQ-v2 [8], a state-of-the-art algorithm for image-based RL. As opposed to the reward-free setting ROT operates in, DrQ-v2 has access to environments rewards. The results in Fig. 5 show that ROT handily outperforms DrQ-v2. This clearly demonstrates the usefulness of imitation learning in domains where expert demonstrations are available over reward-based RL. We also compare against a demo-assisted variant of DrQ-v2 agent using the same pretraining and regularization scheme as ROT (refer to Appendix H.3). Interestingly, we find that our soft Q-filtering based regularization can accelerate learning of RL with task rewards, which can be seen in the high performance of the demo-assisted variant of DrQ-v2.

## 5 Related Work

**Imitation Learning (IL)**    IL [34] refers to the setting where agents learn either from an expert policy or from demonstrations derived from an expert policy without access to environment rewards. IL can be broadly categorized into Behavior Cloning (BC) [1, 21] and Inverse Reinforcement Learning (IRL) [4, 22]. BC solely learns from offline demonstrations but suffers on out-of-distributions samples [5] whereas IRL focuses on learning a robust reward function through online interactions but suffers from sample inefficiency [7]. Deep IRL methods can be further divided into two categories: (1) adversarial learning [35] based methods, and (2) state-matching [36, 37] based methods. GAIL [6] is an adversarial learning based formulation inspired by maximum entropy IRL [38] and GANs [35]. There has been a significant body of work built up on GAIL proposing alternative losses [30, 39, 29], and enhancing its sample efficiency by porting it to an off-policy setting [7]. There have also been visual extensions of these adversarial learning approaches [40, 41, 42]. However, although adversarial methods produce competent policies, they are inefficient due to the non-stationarity associated with iterative reward inference. To alleviate the non-stationary reward problem with adversarial IRL frameworks, a new line of OT-based state-matching approaches have recently been proposed [12, 13, 11].

**Optimal Transport (OT)**    OT [36, 37] is a tool for comparing probability measures while including the geometry of the space. In the context of IL, OT computes an alignment between a set of agent and expert observations using distance metrics such as Sinkhorn [33], Gromov-Wasserstein [43], GDTW [44], CO-OT [45] and Soft-DTW [46]. For many of these distance measures, there is an associated IL algorithm, with SIL [12] using Sinkhorn, PWIL [13] using greedy Wasserstein, GDTW-IL [44] using GDTW, and GWIL [47] using Gromov-Wasserstein. Recent work from Cohen et al. [11] demonstrates that the Sinkhorn distance [12] produces the most efficient learning among the discussed metrics. They further show that SIL is compatible with high-dimensional visual observations and encoded representations. Inspired by this, ROT adopts the Sinkhorn metric for its OT reward computation, and improves upon SIL through adaptive behavior regularization.

**Behavior Regularized Control**    Behavior regularization is a widely used technique in offline RL [48] where explicit constraints are added to the policy improvement update to avoid bootstrapping on out-of-distribution actions [49, 50, 51, 52, 53, 54]. In an online setting with access to environment rewards, prior work [16, 10] has shown that behavior regularization can be used to boost sample efficiency by finetuning a pretrained policy via online interactions. For instance, Jena et al. [17] demonstrates the effectiveness of behavior regularization to enhance sample efficiency in the context of adversarial IL. ROT builds upon this idea by extending to visual observations, OT-based IL, and adaptive regularization, which leads to improved performance (see Appendix H.4). We also note that the idea of using adaptive regularization has been previously explored in RL [24]. However, ROT uses a soft, continuous adaptive scheme, which on initial experiments provided significantly faster learning compared to hard assignments.

## 6 Conclusion and Limitations

In this work, we have proposed a new imitation learning algorithm, ROT, that demonstrates improved performance compared to prior state-of-the-art work on a variety of simulated and robotic domains. However, we recognize a few limitations in this work: (a) Since our OT-based approach aligns agents with demonstrations without task-specific rewards, it relies on the demonstrator being an 'expert'. Extending ROT to suboptimal, noisy and multimodal demonstrations would be an exciting problem to tackle. (b) Performing BC pretraining and BC-based regularization requires access to expert actions, which may not be present in some real-world scenarios particularly when learning from humans. Recent work on using inverse models to infer actions given observational data could alleviate this challenge [55]. (c) On robotic tasks such as *Peg in box (hard)* and *Pressing a switch* from Fig. 4, we find that ROT's performance drops substantially compared to other tasks. This might be due to the lack of visual features corresponding to the task success. For example, in the 'Peg' task, it is visually difficult to discriminate if the peg is in the box or behind the box. Similarly for the 'Switch' task, it is difficult to discern if the button was pressed or not. This limitation can be addressed by integrating more sensory modalities such as additional cameras, and tactile sensors in the observation space.

**Acknowledgments**

We thank Ben Evans, Anthony Chen, Ulyana Piterbarg and Abitha Thankaraj for valuable feedback and discussions. This work was supported by grants from Honda, Amazon, and ONR awards N00014-21-1-2404 and N00014-21-1-2758.

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
