# OpenReview forum: "Watch and Match: Supercharging Imitation with Regularized Optimal Transport"
_robot-learning.org/CoRL/2022/Conference — CoRL 2022 Oral_

### Official Review · Reviewer_fHhv · 2022-07-18

**Originality:** Good
**Technical Quality:** Very Good
**Clarity Of Presentation:** Excellent
**Impact:** 3

**Recommendation:**

Strong Accept: I recommend accepting the paper and will argue for my recommendation even if other reviewers hold a different opinion.

**Summary:**

This paper addresses jointly the issues of (1) sample inefficiency of inverse reinforcement learning (IRL) and (2) brittleness of behavior cloning (BC). To do so, this paper introduces a new imitation learning algorithm, "Regularized Optimal Transport" (ROT). The particular novelty of ROT as I understand it is to initialize a policy via BC, then refine that policy using IRL, with rewards computed using optimal transport (OT). A key insight seems to be to take care during the fine tuning step to (a) ensure that the fine tuning step doesn't take the refined policy "too far" away from the BC policy and (b) to control the degree of regularization with a simple heuristic based on the relative performance of the BC policy compared to the fine-tuned policy at a given iteration. The approach is shown to give good performance across a variety of tasks both in benchmark simulations and on a real robot.

**Issues:**

Recapping from "Weaknesses" above, and giving a little more detail:

- The present experimental evaluation compares the soft Q-filtering approach to hand-tuned regularization. However, given that [24] uses essentially the same idea, but with a hard assignment, it seems like the relevant experimental comparison would be to the hard assignment form of the regularization in eq (4) of this paper. It is mentioned briefly that on some experiments the soft assignment scheme improved speed of policy learning, though no such demonstration is presented (unless I missed something).

An interesting addition to this paper would be an explicit comparison with the hard assignment approach. Given that (as I understand it) this paper is proposing both OT based rewards and an additional soft regularization, I'm not sure this is strictly "necessary" but such a comparison would be interesting and perhaps support the claimed improvements in speed versus the hard assignment approach.

- The novelty with respect to prior work might be clarified a bit. In particular, as I understand it, the novelty is in both the use of the optimal transport reward function and the soft Q-filter regularization. In the context of prior work using similar regularization approaches (i.e. the hand-tuned approaches or hard assignment) then, I think this method differs in both the formulation of the reward *and* in the regularization. Is that correct? I wasn't 100% certain from the paper. I'll try to provide a little more detail in Issues.

Here I can try to be more precise. In Section 3.2, the OT rewards are mentioned, but they don't appear in the actual optimization objective for the policy in eq (3). As I understand it, the Q function is building up an approximation of the value function with rewards computed via optimal transport. Is that correct? If so, it may be worth mentioning this under eq (3). If not, this should definitely be clarified.

**Quality Of The Limitations Section:**

Limitations are addressed clearly

**Reviewer Expertise:**

3: The reviewer is fairly confident that the evaluation is correct

**Robotics Focus:**

Sufficient demonstration on hardware

**Strengths And Weaknesses:**

# Strengths

- I found the paper to be very clear and well-written. I am not an expert in IRL, but despite that I had no trouble following the technical approach. I believe this paper would be very accessible to a reader.

- The overall idea is simple and intuitive; or at least, from the presentation it seems to follow quite naturally.

- The approach seems technically sound.

- The benchmark simulation + real robot experiments were well-presented and thorough overall, but see Weaknesses for a minor comment on this.

- The approach was demonstrated on a real robot.

# Weaknesses

- The present experimental evaluation compares the soft Q-filtering approach to hand-tuned regularization. However, given that [24] uses essentially the same idea, but with a hard assignment, it seems like the relevant experimental comparison would be to the hard assignment form of the regularization in eq (4) of this paper. It is mentioned briefly that on some experiments the soft assignment scheme improved speed of policy learning, though no such demonstration is presented (unless I missed something).

- The novelty with respect to prior work might be clarified a bit. In particular, as I understand it, the novelty is in both the use of the optimal transport reward function and the soft Q-filter regularization. In the context of prior work using similar regularization approaches (i.e. the hand-tuned approaches or hard assignment) then, I think this method differs in both the formulation of the reward *and* in the regularization. Is that correct? I wasn't 100% certain from the paper. I'll try to provide a little more detail in Issues.

## Update after rebuttal

All of my comments have been addressed and I have upgraded my rating from "weak accept" to "strong accept."

**Summary Of Recommendation:**

Overall, I felt this was a very well-written and executed paper. The technical ideas and approach were clear and well-motivated. The main two technical contributions seem to be in the use of optimal transport-based rewards and the "soft Q-filter" regularization when fine-tuning models trained using behavior cloning. The experimental evaluation is thorough and convincing. My main suggestions to improve the work are minor: i.e. to add a comparison to the hard-assignment regularization approach (for completeness), and to perhaps clarify the contribution with respect to existing work.

## Update after rebuttal

I have upgraded my recommendation to "strong accept" as my comments have been addressed.

---

> ### Author Response · Authors · 2022-08-18
> **Response to Official Review of Paper35 by Reviewer fHhv**
>
> **Comment:**
>
> Thank you for your thoughtful comments on the paper. We are glad that you found the paper well-written and executed. We provide clarifications for your questions below.
>
> - **Comparison between soft Q-filtering and hard assignment**: We have provided an explicit comparison between our proposed soft Q-filtering approach and the hard assignment proposed by Nair et al. [1].  The comparison has been provided in the appendix in Section H.4 under the title “Choice of Q-filtering method”. We observe that though the two strategies have comparable asymptotic performance in some cases, soft Q-filtering exhibits better sample efficiency and more stable training.
> - **Clarification about the use of OT rewards in the IRL framework**: In this work, we use inverse reinforcement learning (IRL) for the online learning phase. Keeping in line with the approach adopted by previous trajectory-matching based IRL algorithms, we compute the reward for each step of an online trajectory rollout by matching it to the demonstration trajectory(s) using an optimal transport based reward computation. This reward is then plugged into a DrQ-v2 style RL framework to optimize the agent neural networks. Thus, like you mentioned in your comment, the OT reward does not show up in Equation 3 because the Q-function approximates the value function and the Q-function is optimized using the OT-rewards. To avoid confusion, we have made a revision of the paper that explicitly mentions this.
>
> We hope we have been able to address your questions in the above clarifications. Kindly let us know if you have additional questions and we would be more than happy to discuss further.
>
>
> **Zip File:**
>
> /attachment/b3f0bd06728db19d0e2802618ceadc73a90bbff8.zip

---

> > ### Comment · Reviewer_fHhv · 2022-08-26
> > **Response to Paper35 Authors**
> >
> > Thank you for your detailed responses - all of my comments have been addressed.

---

### Official Review · Reviewer_xvuE · 2022-08-01

**Originality:** Very Good
**Technical Quality:** Excellent
**Clarity Of Presentation:** Excellent
**Impact:** 4

**Recommendation:**

Strong Accept: I recommend accepting the paper and will argue for my recommendation even if other reviewers hold a different opinion.

**Summary:**

Summary: This paper proposes Regularized Optimal Transport, a new imitation learning algorithm that builds on top of optimal transport-based trajectory matching. The algorithm can learn from a few expert demonstrations and an online reinforcement learning phase. The algorithm is benchmarked in simulation and on real robots and achieved a high success rate.


**Issues:**

No major issues but hope to get clarification of the questions mentioned in the review.

**Quality Of The Limitations Section:**

Limitations are addressed clearly

**Reviewer Expertise:**

4: The reviewer is confident but not absolutely certain that the evaluation is correct

**Robotics Focus:**

Sufficient demonstration on hardware

**Strengths And Weaknesses:**

### Strengths

1. This is a great paper with solid work put in. It proposes an effective imitation learning algorithm that has strong performance. The proposed method works well on both simulated benchmarks and on real robots. On simulated benchmarks, it is several times faster in sample efficiency. And on real robotics tasks, the proposed method can achieve 90% success rate with 1 demonstration and 1 hour of online learning.
2. The authors provide a detailed analysis of the design choices and contribution of each component, providing insight of why soft Q-filtering and OT-based rewards are needed to stabilize the training.
3. Despite not in the main text, Figure 6 is very helpful to understand the intuition of the proposed method and why it is promising.

### Weaknesses

1. The authors showed that the algorithm is somewhat robust to the out-of-distribution initial state. I am wondering how robust the proposed method is to other types of out-of-distribution states, such as adding a task-irrelevant distractor object in the scene during the second stage of training.
2. What action space does the real robot experiment use and does the proposed method work with other types of action spaces, e.g. impedance control, for contact-rich tasks, such as wiping a board?
3. What is the extent of initial condition deviation the proposed method can handle? When the deviation is too big, would there be a scheme where not initializing the policy with $\pi_{BC}$ be more beneficial?

**Summary Of Recommendation:**

Given the strong results and the thoroughness of the experiments, answers to scientific questions proposed in the experiment section, and the clarity in writing, I recommend a strong acceptance for this paper. I believe this is a great paper to have for the community.

---

> ### Author Response · Authors · 2022-08-18
> **Response to Official Review of Paper35 by Reviewer xvuE**
>
> We thank you for your insightful comments about the paper and we are glad that you found our work convincing and thorough. We provide clarifications to your questions below.
>
> - **Clarification regarding performance of ROT on out-of-distribution states**: In this work, we primarily focus on adaptively combining trajectory-matching rewards with behavior cloning to accelerate imitation learning. Regarding out-of-distribution examples, our second stage of training involves robot and object starting states that were not seen in the single expert demonstration. You are correct that we do not explicitly evaluate with distractor objects. This is indeed an interesting and important problem that we would like to extend ROT to in the future. We conjecture that ROT will require an explicit mechanism to learn representations that are agnostic to distractors such as self-supervision on the data observed in the second phase of training.
> - **Clarification regarding the action space for real-world experiments**: Our real-world experiments are done using the XArm 7 robot, which is a position-controlled robot with its action space consisting of relative gripper movement in the 3D space and relative movement (opening and closing) of the gripper fingers. The proposed algorithm is invariant to the structure of the action space and hence should work on other action spaces as well. For instance in the DM Control suite, the action space is joint torques.
> - **Regarding the extent of initial condition deviation that ROT can handle**: As you mentioned in your comment, Figure 6 in the appendix demonstrates the ability of ROT to handle moderate deviations from the initial condition shown in the demonstration. For the real-world experiment and the Metaworld tasks, which had access to only a single demonstration trajectory, we observe that even for the maximum possible deviation in the initial condition, the model learns to match its trajectory to the demonstration trajectory towards the later part of the rollout in order to achieve the goal. Additionally, based on our observations, since the online-learning stage starts from a behavior cloned initialization, it starts performing well when it encounters a state from the demonstration dataset. This also illustrates why adaptively weighting the BC objective with the RL objective is important. Intuitively, for out-of-distribution states, we would like the robot to focus more on the RL objective, while on in-distribution states to focus on the BC objective. Without the BC initialization, the policy would have to train from scratch (similar to the OT baseline in the paper) which needs significantly more samples to achieve the same performance as ROT.
>
> We hope we have been able to address your questions in the above clarifications. Kindly let us know if you have additional questions and we would be more than happy to discuss further.

---

### Official Review · Reviewer_qDWt · 2022-08-01

**Originality:** Very Good
**Technical Quality:** Very Good
**Clarity Of Presentation:** Very Good
**Impact:** 4

**Recommendation:**

Strong Accept: I recommend accepting the paper and will argue for my recommendation even if other reviewers hold a different opinion.

**Summary:**

This paper attempts to accelerate the imitation with a few demonstrations. It adaptively combines offline behavior cloning and online trajectory-matching based rewards. Specifically, the paper contributes a method of adaptive regularization with soft Q-filtering, which regularizes the IRL policy to stay close to the BC pretrained policy. The paper shows in various tasks that the proposed method can accelerate the imitation while at the same time reach high success rate.

**Issues:**

1.	As mentioned in “Weaknesses”, the paper mentions that this performance drop might be due to the lack of visual features, but for fetch_push, fetch_reach, fetch_pick_and_place, the performance is also unstable and worse than other baselines. Since tasks like reach and push do not need much visual features, what could be reason for that?

2.	As mentioned in the paper, the expert policy is trained using DrQ-v2 for DeepMind Control tasks and OpenAI Robotics tasks. While in section 4.5, the proposed ROT is compared and outperforms DrQ-v2. Could you clarify what’s the difference between the expert policy DrQ-v2 and the one is compared?

3.	In figure4, the metaworld_hammer plot is missing the line for expert.


**Quality Of The Limitations Section:**

Limitations are addressed clearly

**Reviewer Expertise:**

4: The reviewer is confident but not absolutely certain that the evaluation is correct

**Robotics Focus:**

Sufficient demonstration on hardware

**Strengths And Weaknesses:**

Strengths:

•	The paper is well motivated in such that it proposes improving the efficiency of imitation learning both by decreasing the training time and cost of demonstrations.

•	The experimental evaluation is extensive. The authors experimented on 20 simulation tasks and 14 real robot tasks and provides convincing results that the proposed method generally outperforms baselines in these tasks.

•	The figures in the paper are well designed and composed. The ablation experiments are detailed and clear.

Weaknesses:

•	As the paper admits, on tasks such as Peg in box (hard) and Pressing a switch, ROT’s performance drops substantially. Also in some other tasks, the performance is not stable.


**Summary Of Recommendation:**

The paper proposes to accelerate the imitation with a few demonstrations. The paper is well-written and overall technical sound. The literature review is thorough and experiment results are extensive and convincing.

---

> ### Author Response · Authors · 2022-08-18
> **Response to Official Review of Paper35 by Reviewer qDWt**
>
> **Comment:**
>
> We thank you for your thoughtful review of the paper and for finding our work well motivated and extensively evaluated. We address your questions and provide clarifications below.
>
> - **Concerns regarding performance drop due to lack of visual features**: In Figure 2 of the paper, we demonstrate the performance of ROT compared to other baselines on a set of pixel-based continuous control tasks. This shows that ROT outperforms other baselines in fetch_reach and fetch_pick_and_place and is slightly worse than DAC for fetch_push. Although the performance is unstable, it is relatively more stable as compared to the other IRL baselines (shown by the standard deviation of the curves). We conjecture that unstable training for tasks like fetch_push and fetch_pick_and_place results from a lack of visual features since the block only spans a few pixels in the 84x84 sized frames passed through the encoder network. One possible fix for this issue is to use egocentric camera views that contain a substantially larger view of the robot’s workspace.
> - **Clarification about training of the expert policy**: We use image-based DrQ-v2 for collecting expert demonstrations for the DeepMind Control suite. However, for the OpenAI Robotics tasks, we use a state-based DrQ-v2 with Hindsight Experience Replay (HER)  [1] as our expert demonstrations. In Section 4.5 and consequently in Figure 4, DrQ-v2 (RL) refers to an image-based DrQ-v2 trained using environment rewards on the corresponding tasks. As a result, we observe in Figure 4 that over the training cycle, the DrQ-v2 policy (though less sample efficient than ROT), reaches the same reward value as the expert policy for the tasks in DeepMind Control Suite. However, image-based DrQ-v2 without HER proves to be inadequate to solve the OpenAI Robotics tasks which is evident from the performance in Figure 4. Our choice of using DrQ-v2 without HER is informed by (a) making sure our comparisons are apples-to-apples across all methods and (b) training image-based HER without state information is quite challenging since it requires goal identification from images.
> - **Missing line for expert in the metaworld_hammer plot**: We thank you for bringing this to our attention and we have fixed the plot in the revised version of the paper.
>
> We hope we have been able to address your concerns in the above clarifications. Kindly let us know if you have additional questions and we would be more than happy to discuss further.
>
> [1] Andrychowicz, Marcin, et al. "Hindsight experience replay." Advances in neural information processing systems 30 (2017).
>
>
> **Zip File:**
>
> /attachment/cceeb27a9075c9e77bb5e255eb9c6a0925815dc7.zip

---

> > ### Comment · Reviewer_qDWt · 2022-08-26
> > **Thank you for the response**
> >
> > I thank the authors for their clear reply. All my concerns have been addressed

---

### Official Review · Reviewer_Ldfy · 2022-08-01

**Originality:** Very Good
**Technical Quality:** Excellent
**Clarity Of Presentation:** Excellent
**Impact:** 4

**Recommendation:**

Strong Accept: I recommend accepting the paper and will argue for my recommendation even if other reviewers hold a different opinion.

**Summary:**

The paper proposes a method for robustifying a policy obtained via behavior cloning by collecting additional experiences in the environment and minimizing an optimal-transport-based (OT-based) trajectory matching reward between expert demonstrations and trajectories generated by the behavior policy (Algorithm 1 in Appendix C). Additionally, the paper proposes a particular adaptive scheme (Eq. (4)) for tuning the weight between the objectives for behavior cloning and for maximizing the OT-based returrn. Experiments on vision-based control tasks from DeepMind Control suite, OpenAI Robotics suite, and Meta-World suite, as well as real-world experriments on an xArm robot arm demonstrate that the proposed method ROT compares favorably to Discriminator Actor Critic (DAC) [7], even when DAC is augmented with the proposed prertreaining and regularization scheme, and furthermore ROT outperforms a state of the art OT-based Sinkhorn Imitation Learning (SIL) algorithm [12].

**Issues:**

None

**Quality Of The Limitations Section:**

Limitations are addressed clearly

**Reviewer Expertise:**

4: The reviewer is confident but not absolutely certain that the evaluation is correct

**Robotics Focus:**

Sufficient demonstration on hardware

**Strengths And Weaknesses:**

Strengths
- The paper provides a strong empirical evaluation and convincing results for the proposed ROT method
- The presentation is clear, the questions addressed by the experiments are explicit and easy to follow
- The method combines several ideas from prior works (as well explained in Sec. 5) but is sufficiently novel and may become a new baseline for BC+finetuning algorithms, plus further extensions are possible

Weaknesses
- A few important details about the algorithm can only be found in the appendix (e.g., what networks exactly are trained, feature preprocessing for OT, freezing of the feature processor, etc.). However this is OK due to the space limitations.

**Summary Of Recommendation:**

The paper proposes a novel imitation learning algorithm that adaptively combines offline behavior cloning with online trajectory-matching based rewards. The method is thoroughly evaluated and shown to outperform the baselines, including in experiments on real hardware. There are no concerns with regards to quality, significance, clarity, etc.

---

> ### Author Response · Authors · 2022-08-18
> **Response to Official Review of Paper35 by Reviewer Ldfy**
>
> Thank you for your positive response to the work. We are glad that you find the work interesting and easy to follow. We agree that it would be nice to have additional details pertaining to network architecture, training details, ablations, etc. in the main paper. However, as you mentioned in your comments, space constraints have led us to push these details to the appendix in order to focus more on the experimental results and applicability to robotics in the main paper.

---

### Author Response · Authors · 2022-08-18
**Global comments and revision summary in response to the reviews of Paper35**

**Comment:**

We thank the reviewers for their insightful and constructive comments. We are glad that this work was received positively, although some important clarifications were asked for. We have provided these clarifications as a detailed response to each individual review and provide a summary of the revisions made below.

- **Fixed missing expert line in Figure 4**: As per Reviewer qDWt’s comments, we have added the missing expert line in the metaworld_hammer plot in Figure 4.
- **Comparison between soft Q-filtering and hard assignment**: As per reviewer fHhv’s comments, we have provided a comparison between our proposed soft Q-filtering approach and the hard assignment proposed by Nair et al. [1]. Owing to space constraints in the main paper, the plots for the comparison have been added to the appendix in Section H.4 under the title “Choice of Q-filtering method”. The plots highlight the better performance of soft Q-filtering with regard to sample efficiency and training stability.
- **Clarification about the use of OT rewards in the IRL framework**: As per reviewer fHhv’s comments, we have made an explicit mention of the OT-rewards being used for the optimization of  the critic responsible for Q-value estimation. The addition has been made right after Equation 3 and we hope to avoid any confusion on the reader’s part through this.

We hope that these updates to our paper inspire further confidence in our work. At the same time, we invite any further questions or feedback that you may have on our work.

[1] A. Nair, B. McGrew, M. Andrychowicz, W. Zaremba, and P. Abbeel. Overcoming exploration in reinforcement learning with demonstrations. In 2018 IEEE international conference on robotics and automation (ICRA), pages 6292–6299. IEEE, 2018.

**Zip File:**

/attachment/d7cdd26c6ee27e231877f774f69224fa10dc391f.zip

---

### Meta-Review · Area_Chair_JN41 · 2022-08-09

**Recommendation:** Accept (Oral)
**Confidence:** 5

**Metareview:**

This paper proposes a new imitation learning algorithm, Regularized Optimal Transport, which adaptively combines trajectory-matching rewards and behavior cloning. The paper demonstrates much faster learning with fewer demonstrations in both simulation and real-world robot learning tasks.

All reviewers agree that the paper is well written, the approach is sound, the evaluations are thorough and the results are strong. There were some questions about the performance drop on several experiments, novelty, and comparison with the hard assignment approach. After the rebuttal and discussions, all of these concerns were addressed. Thus, we would like to recommend accepting this paper.

**Best Paper Nomination:**

Yes